# Inflammasome Activation-Induced Hypercoagulopathy: Impact on Cardiovascular Dysfunction Triggered in COVID-19 Patients

**DOI:** 10.3390/cells10040916

**Published:** 2021-04-16

**Authors:** Lealem Gedefaw, Sami Ullah, Polly H. M. Leung, Yin Cai, Shea-Ping Yip, Chien-Ling Huang

**Affiliations:** Department of Health Technology and Informatics, The Hong Kong Polytechnic University, Kowloon, Hong Kong, China; lealem.bimerew@connect.polyu.hk (L.G.); samikhan.ullah@polyu.edu.hk (S.U.); polly.hm.leung@polyu.edu.hk (P.H.M.L.); david-yin.cai@polyu.edu.hk (Y.C.)

**Keywords:** inflammation, hypercoagulopathy, cardiovascular complications, COVID-19, NLRP3

## Abstract

Coronavirus disease 2019 (COVID-19) is the most devastating infectious disease in the 21st century with more than 2 million lives lost in less than a year. The activation of inflammasome in the host infected by SARS-CoV-2 is highly related to cytokine storm and hypercoagulopathy, which significantly contribute to the poor prognosis of COVID-19 patients. Even though many studies have shown the host defense mechanism induced by inflammasome against various viral infections, mechanistic interactions leading to downstream cellular responses and pathogenesis in COVID-19 remain unclear. The SARS-CoV-2 infection has been associated with numerous cardiovascular disorders including acute myocardial injury, myocarditis, arrhythmias, and venous thromboembolism. The inflammatory response triggered by the activation of NLRP3 inflammasome under certain cardiovascular conditions resulted in hyperinflammation or the modulation of angiotensin-converting enzyme 2 signaling pathways. Perturbations of several target cells and tissues have been described in inflammasome activation, including pneumocytes, macrophages, endothelial cells, and dendritic cells. The interplay between inflammasome activation and hypercoagulopathy in COVID-19 patients is an emerging area to be further addressed. Targeted therapeutics to suppress inflammasome activation may have a positive effect on the reduction of hyperinflammation-induced hypercoagulopathy and cardiovascular disorders occurring as COVID-19 complications.

## 1. Introduction

Host immune response to microbes and complex diseases is a critical systemic defense process of the human body. The innate immune system is the first line of defense that identifies stimuli experienced by the host. It involves immune cells and recognizes foreign agents through different pattern recognition receptors (PRRs), including toll-like receptors (TLRs), C-type lectin receptors, RIG-like helicase (RLR), cytosolic DNA sensors, and members of the nucleotide-binding oligomerization (NOD)-like receptor (NLR) family [1,2]. Containing a NOD domain, a leucine-rich repeat (LRR) domain, and a pyrin domain, the NLR family pyrin domain containing 3 (NLRP3) protein is the most extensively characterized NLR to date, although there are numerous NLRs identified so far [3,4]. Inflammasomes are large multiprotein intracellular complexes that play a central role in the innate immune response of the host [5]. Inflammasome complexes are formed in the cytoplasm of innate immune cells mainly of the myeloid lineages, such as macrophages and dendritic cells, in reaction to pathogen-associated molecular patterns (PAMPs) or damage-associated molecular patterns (DAMPs). Numerous inflammasome forms have been identified so far that comprise a sensor molecule, an adaptor molecule known as ASC (apoptosis-associated speck-like protein containing a caspase activation and recruitment domain (CARD)), and pro-caspase-1 [1].

Activation of the inflammasome is a crucial step in the regulation of an innate immune response. The process of inflammasome activation is caspase-dependent, involving canonical or non-canonical immune regulation of different caspase types. The canonical pathway is caspase-1 dependent. Caspase-1 converts proinflammatory cytokines into their functional counterparts (mainly pro-IL-1β and pro-IL-18 into IL-1β and IL-18, respectively). Following activation, these proinflammatory cytokines induce inflammation in the host, as part of the response, against microorganisms or foreign substances. Active caspase-1 additionally triggers pyroptosis, an inflammatory form of cell death by which host immune cells eliminate foreign invaders [1,3,6]. On the other hand, the non-canonical pathway involves caspase-4 and caspase-5, which directly trigger pyroptosis. The involvement of inflammasomes in multiple viral, bacterial, and parasitic infections is a novel target area of research on the molecular mechanisms underlying diseases. Several reports have confirmed the immunomodulatory roles of different inflammasomes in multiple viral infections [7,8].

COVID-19 patients develop acute respiratory syndrome characterized by inflammation and pneumonia. The virus can be acquired in a number of ways. Oral saliva is also a source of SARS-CoV-2, which may aggravate the problem in individuals with poor oral hygiene, including in children [9,10,11]. In severe cases (mainly in adults), uncontrolled cytokine activity such as IL-1β, IL-6, and IL-8 contributes to excessive inflammatory responses that can lead to ‘cytokine storm’ syndrome [12,13]. Inflammasomes are suggested to play an active role in COVID-19 disease pathogenesis [14]. In view of the emerging regulatory functions of these inflammasomes in thrombosis, the related molecular mechanisms and signaling pathways, the behavior of inflammasomes in COVID-19 pathogenesis, the mechanisms underlying hypercoagulopathy, and the potential molecular targets, particularly for cardiovascular disorders, are discussed in detail in this article.

## 2. Molecular Basis of Inflammasome Activation

In both the afferent and efferent arms of the immune response, macrophages are significantly involved in the ingestion of foreign microorganisms (PAMPs) or endogenous compounds (DAMPs) such as asbestos crystals, adenosine triphosphate (ATP), and uric acid. Through complementary PRRs, macrophages initially recognize foreign or endogenous substances. This process of detection assists in phagocytosis. PRRs are in contact with the environment external to macrophages and dendritic cells, and can identify PAMPs. Degraded lysosomes are released after absorption. The release of lysosomal content is considered a critical mechanism for activation of inflammasomes although the potential direct link between inflammasome activation and lysosome release requires further investigation [15,16].

The majority of inflammasomes characterized to date contain receptor sensor molecules, which include NLR, NLRP1, NLRP3, NLRP6, NLRP12, and NLR family CARD domain containing 4 (NLRC4). Other families of inflammasomes containing an absent in melanoma 2 (AIM2) protein, interferon gamma inducible protein 16 (IFI16), and pyrin have additionally been identified. AIM2 is a member of the PYHIN family that recognizes double-stranded cytoplasmic DNA (dsDNA) [1,3]. This PYHIN family protein is the only inflammatory sensor that does not belong to the NLR family, despite sharing a number of structural features. It is characterized by N-terminal pyrin domain (PYD) and a C-terminal human interferon-inducible nuclear protein with a 200-amino-acid repeat (HIN200) DNA-binding domain [17].

A typical inflammasome complex comprises molecules or receptors of the inflammasome sensor, adaptor protein ASC, and caspase-1. Inflammasome activation may involve multiple potential mechanisms, including phagolysosomal destabilization, generation of reactive oxygen species (ROS), and induction of transmembrane ion fluxes, such as K^+^ and Ca^2+^ (Figure 1) [15]. In macrophages, endogenous stimulation via DAMPs, such as crystals, ATP, and nigericin, triggers ROS production, indirectly causing inflammatory pathway activation. In response to DAMP recognition, NLRs undergo ATP-dependent oligomerization and enlist ASC through PYD–PYD interactions. Pro caspase-1 is subsequently recruited through ASC CARD, which is essential for its activation [18]. Typically, ASC is localized in the nucleus of immune cells and redistributed to the cytosol in the presence of NLRs and caspase-1 during infection. As nuclear retention of ASC ultimately impairs inflammasome-mediated IL-1β release, this step is necessary for the aggregate formation and stimulates inflammasome activity. To support the formation of caspase-1 dimer, ASC monomers are required to induce proximity-mediated autoactivation of caspase-1 [8].

Caspases are a family of cysteine proteases specific for aspartate, and act as primary mediators of inflammation and apoptosis. Caspases are synthesized as inactive zymogens and possess a variable-length pro-domain, followed by large (20 kDa) and small (10 kDa) subunit active domains. Apoptotic initiator caspases (caspase-2, -8, -9, and -10), apoptotic effector caspases (caspase-3, -6, and -7), and caspases involved in inflammatory cytokine synthesis (caspase-1, -4, -5, and -12L/12S) are subdivided into three functional classes [19]. Among these molecules, caspase-1 is the most extensively characterized inflammatory caspase and significantly implicated in the innate immune response.

Activation of caspase-1 is necessary for maturation and release of proinflammatory cytokines, primarily pro-IL-1β and pro-IL-18. Pro-IL-1β is expressed at low levels in resting cells and requires activation by specific factors such as PRRs. Proinflammatory IL-1β is an acute-phase response mediator that promotes immune cell inflammation, vasodilation, hyperthermia, and extravasation [15]. IL-1β, also identified as an endogenous pyrogen, is involved in cell proliferation, differentiation, and development [1]. IL-18 (interferon gamma inducing factor) is an IL-18 gene-induced proinflammatory cytokine. The IL-18 precursor is present in virtually all healthy human and animal cells, unlike IL-1β. An elevated level of IL-18 is correlated with the severity of the disease [20]. In the presence of IL-12, it is found to activate Th1 cells to produce IFN-γ. IL-18 not only stimulates NK cells, basophils, and mast cells, but it also plays a role in inflammation [21].

Caspase-1 can also cleave gasdermin D (GSDMD) protein, inducing a type of cell death known as pyroptosis. Cell death occurs as a result of membranous pore formation, cytoplasmic swelling, and leakage of cytosolic content. Pyroptosis is caused by proinflammatory signals and associated with inflammation, presenting a programmed caspase-1-dependent means by which host cells clean up intracellular microorganisms [15]. Cells undergoing pyroptosis release elevated levels of IL-1β and IL-18, and display DNA fragmentation along with nuclear condensation [22]. AIM2 sensor proteins can also recognize the release of dsDNA from pyroptotic cells. The ASC will be recruited through PYRIN–PYRIN interaction through these sensing processes. The ASC intern can interact with caspase-1 via its CARD domain, and activate caspase-1 [18]. AIM2 inflammasome regulates host immune response by interacting with cytosolic dsDNA from bacteria and viruses [23].

In addition to NLRP3, other NLRs are being investigated for the regulation of the innate immune response and their potential advantage as therapeutic targets in different inflammatory conditions. The NLRP2 inflammasome is another multiprotein complex comprising NLRP2, the adaptor protein ASC, and caspase-1 that interacts with the P2X7 receptor and pannexin-1 channel. Stimulation of human astrocytes with ATP results in activation of the NLRP2 inflammasome, leading to the processing of inflammatory caspase-1 and IL-1β [24].

## 3. Mechanisms of Blood Clotting Induced by Activated Inflammasome

Blood clot formation is the mechanism by which blood changes from a liquid to a gel state. Coagulation progresses almost immediately after an injury occurs to the endothelial lining of blood vessels [25]. Coagulation and inflammation are intimately interlinked, and dysregulation of single components of such systems may impact the entire equilibrium, resulting in a broad range of diseases that involve various forms of increased inflammation and thrombosis. Inflammation initiates coagulation, suppresses the functional pathways of natural anticoagulants, and impairs the fibrinolytic system. Inflammatory mediators can elevate platelet count and reactivity, downregulate natural anticoagulant activities, activate the coagulation system, promote the progression of coagulant response, and impede fibrinolysis [26,27]. Similarly, clotting induces an increase in the inflammatory response by releasing mediators from platelets and activated cells, thereby facilitating cell–cell interactions that increase inflammatory responses [28,29].

Inflammasome-induced blood clotting may occur because of several interrelated mechanisms including damage to the blood vessels, hypoxia, platelet activation, and proinflammatory cytokines. Endothelial cells (ECs) perform a range of critical roles in regulating vascular functions [25]. Under various conditions, such as pathological inflammatory and thrombotic stimuli, microparticles (MPs) are released from activated ECs, which may retain some RNA and cytosolic material. These EC damage derivatives are involved in inflammation, hemostasis, and the control of neutrophil extracellular trap (NET). An earlier study reported that monocytic MPs activate ECs through NLRP3 inflammasome-mediated activation, in turn inducing ERK1/2 phosphorylation, nuclear factor-kappa B (NF-kB) pathway activation and cell adhesion molecule expression, intercellular adhesion molecule 1, vascular cell adhesion molecule-1, and E-selectin [30]. These MPs contain tissue factor (TF) and induce clot formation by interacting with factor VIIa, the primary triggering factor in the extrinsic blood clot pathway [31,32].

A close correlation between hypoxia and inflammation has been highlighted recently, involving activation of multiple cell types such as lymphocytes, platelets, and endothelium. Hypoxia refers to a condition in which the body or a region of the body is starved of sufficient oxygen supply at the tissue level. Hypoxia induces hypoxia-inducible factor 1-alpha (HIF-1α) to be engaged in the hypoxia-induced expression of NLRP3 and thrombosis. In an animal model, treatment of hypoxia-mediated thrombosis using small interfering RNA (siRNA) targeting HIF-1α decreased transcription of NLRP3, IL-1β, and caspase-1 [25] through the recruitment of neutrophils and macrophages and the release of lytic enzymes such as myeloperoxidase and chemotactic factors that increase NETs. These NETs, consisting of DNA and histones, are released by leukocytes in a process known as NETosis. Released histones and DNA molecules, in turn, activate platelets and increase thrombosis via stimulating the main platelet adhesion receptor integrin αIIbβ3, phosphatidylserine exposure, FV/Va expression, and thrombin development [32].

Platelets are small non-nucleated cell fragments circulating in the blood and play a significant role in managing vascular integrity and hemostasis. NLRP3 does not affect platelet production, platelet receptor expression, or granule expression. However, platelet NLRP3 deficiency significantly impairs hemostasis and in vivo formation of arterial thrombus [33]. NLRP3 regulates the spread of platelets and retraction of clots via an IL-1β-dependent mechanism. Moreover, impaired hemostasis and arterial thrombosis have been reported in mice containing NLRP3-deletion platelets. NLRP3 inhibition impairs the retraction of clots in human platelets [34]. In a murine model of pancreatic cancer, the platelet NLRP3 inflammasome is upregulated and promotes platelet aggregation and tumor formation [35]. Platelet activation triggers the release of intragranular contents like ATP. NLRP3 binds intracellular DAMPs, such as ATP, leading to activation of inflammasome pathways. Release of IL-1β in platelet MPs is also dependent on caspase 1, based on the finding that inhibition of caspase-1 results in a decrease in IL-1β particle release [33,36].

Interleukin-6 (IL-6) is a multifunctional cytokine that plays a critical role in various biological processes, not just the immune system. This cytokine is a crucial regulator for both chronic and acute inflammation [37]. IL-6 stimulates TF to transform prothrombin into thrombin, converting fibrinogen into fibrin. Thus, a significant association is evident between the formation of the inflammasome and the formation of the clot. Furthermore, inflammatory activation causes the release of TF-containing microvesicles via pyroptosis, resulting in systemic coagulation and death [38].

Another mechanism underlying inflammasome-driven blood clotting is induction of pyroptosis [32]. Inflammasome activation results in immune cell activation, disruption of internal cell contents, and damage to the cell membrane. Pyroptosis, a necrotic cell death modality of macrophages, facilitates the release of membrane components, such as TF-MPs, into the circulation. The release of TFs is a prerequisite for activation of coagulation factors in the clotting process [38]. In contrast, lytic cell death activates a novel approach of histone-induced coagulation and thrombosis independent of caspase 1/11 and GSDMD [32]. Histones are cationic nuclear proteins essential for eukaryotic chromatin structure and function. Incubation of histones with macrophages leads to the induction of lytic cell death and phosphatidylserine exposure, the main coagulation initiator necessary for TF activity. Neutralization of TF is reported to reduce coagulation caused by histones [32,39]. Limited studies to date have attempted to evaluate the involvement of inflammasome activation in the thrombotic process, and further research is warranted to elucidate the pathways and therapeutic targets of thrombosis.

## 4. Inflammasome Activation in COVID-19 Patients

The inflammasome has been identified as a bridge between thrombosis and inflammation. Numerous clinical studies indicate that the commonly observed cytokine storm in COVID-19 infection is characterized by the development of inflammasomes. Among the NLRPs, an association of the NLRP3 inflammasome with COVID-19 is reported. Elevated inflammatory marker expression in leukocytes has been demonstrated in patients that have died from COVID-19. Incubation of viable SARS-CoV-2 viral particle with monocytes results in the activation of NLRP3 inflammasome as demonstrated by puncta formation, a marker for active inflammasome formation. Activation of NLRP3 inflammasome is associated with disease severity in COVID-19 patients [40]. In addition, the presence of fatal NLRP3 inflammasome aggregates of COVID-19-induced pneumonia in lung supports the existence of biological relations between viral infection and cytokine release syndrome [41].

Studies have shown that inflammasome activation is triggered by coronavirus structural and accessory proteins. Viroporins (≤100 amino acid residues) are small virally encoded ion channels that oligomerize in the membrane of host cells, leading to the formation of hydrophilic pores. These proteins play critical roles in viral replication and pathogenesis, and contribute to the transport of Ca^2+^ ions into the cell cytosol. NLRP3 is activated under conditions of high cytosolic Ca^2+^ [42,43]. Spike protein lacking the transmembrane domain activates the inflammasome in macrophages derived from peripheral blood mononuclear cells (PBMCs) in COVID-19 patients. NLRP3 inhibition with a selective MCC950 suppressor results in the reduction of IL-1β secretion from spike protein-stimulated macrophages [44].

Moreover, it has been shown that SARS-CoV orf8b protein could activate the NLRP3 inflammasome in the in vitro setting. In macrophages, orf8b triggers intracellular aggregates, lysosomal stress, autophagy, and pyroptotic cell death [45]. SARS-CoV-2 orf3a protein is formed in a similar manner, and the inflammasome is activated by K^+^ ion efflux and kinase NEK-7. Orf3a-activated IL-1β expression occurs via NF-kB pathway. These leads to disruption of the intracellular ion balance, promoting mitochondrial damage, and generating ROS, which act as co-activators of NLRP3 [46].

The NLRP3 inflammasome also plays a regulatory role in platelet function. The key role of platelets in blood clot formation is primary hemostasis, which involves adherence to damaged surfaces, binding to procoagulants, and forming aggregates to avoid blood loss. NLRP3 in platelets is upregulated with increased caspase-1 activity in pancreatic cancer, as demonstrated in a mouse model [35]. The decrease in platelet activation and aggregation can be induced via the downregulation of NLRP3 [35]. Platelet inflammasome NLRP3 activation is reported to trigger platelet aggregation, endothelial dysfunction, and thrombosis, which may serve as contributing factors to hypercoagulopathy, an issue that requires further investigation in COVID-19 patients.

Infection with SARS-CoV-2 induces cell death characterized by loss of integrity of the plasma membrane, characteristic of pyroptosis. Evaluation of COVID-19 patients and post-mortem samples demonstrated that SARS-CoV-2 induces inflammasome activation in primary human monocytic cells and mimics the release of lactate dehydrogenase, a marker of cell injury, from infected monocytes. According to recent reports, SARS-CoV-2 directly infects human monocytic cells and promotes activation of NLRP3 and lytic cell death [47,48]. Although more evidence is needed to elucidate the role of both structural and non-structural SARS-CoV-2 proteins, understanding the underlying molecular mechanisms could pave the way for a therapeutic target to reduce disease severity.

## 5. Hypercoagulopathy Associated with SARS-CoV-2 Infection

Hypercoagulopathy is a common phenomenon in COVID-19 patients and is associated with illness severity [49]. A review report showed that 209 (69.0%) out of 303 patients had coagulation abnormalities. The most common was an alteration in fibrinogen and D-dimer levels, prolonged prothrombin time (PT), and prolonged activated partial thromboplastin time (APTT) [50]. The amount of D-dimer was higher in patients with severe than mild infection in a related study performed at Wuhan First Hospital, China [51]. Similarly, mean levels of fibrinogen, D-dimer, and von Willebrand factor (VWF) were shown to be significantly increased in a study conducted in Milan, Italy, on COVID-19 patients in the intensive care unit (ICU). PT and APTT were normal or slightly elevated. On the other hand, the antithrombin level was slightly decreased [51]. The levels of PT and D-dimer were significantly higher in non-survivors than survivors of COVID-19 [52].

SARS-CoV-2 infects blood vessels and causes vascular damage, both in vitro and in vivo, characterized by increased procoagulant factors associated with poor prognosis and higher mortality [53]. According to a report by Tang et al., 71.4% of COVID-19-related deaths were related to altered coagulation profiles [54]. Research by Klok and coworkers revealed that the composite incidence of thrombotic events was 31% in COVID-19 patients in ICU, with venous thromboembolism detected in most COVID-19 cases (27% of patients with thrombotic events) [55]. The severity of both macro-and micro-thrombosis was increased in critically ill patients [56]. Similarly, the prevalence of alveolar-capillary microthrombi was another frequently reported phenomenon [49]. In COVID-19 patients with chronic non-communicable diseases, the degree of thrombotic complications is significant since 20–30% of critically ill patients with secondary problems during viral infection are at higher risk of developing thrombotic complications [56,57]. Besides, the associated endothelial damage promotes the recruitment of immune cells, leading in turn to the release of proinflammatory cytokines, acute-phase reactants, ultra-large VWF multimers involved in primary hemostasis and TF overexpression [55,57]. Unnecessary release of proinflammatory cytokines (cytokine storm) and release of other acute-phase reactants can activate the complement system, which induces the intrinsic and extrinsic coagulation cascades, resulting in a state of hypercoagulability [58].

At the cellular level of blood platelets, a study of COVID-19 patients showed increased mean platelet volume and platelet hyperactivity associated with a decrease in platelet count [59,60]. Thrombocytopenia is frequently reported in association with increased risk of severe illness [61,62]. Nevertheless, the causes of thrombocytopenia and platelets’ involvement are not well known. They may be attributed to various processes including inflammation, oxygen demand injury, and plaque rupture caused by the infection. A recent study on platelet gene expression offers new evidence of altered platelet gene expression and substantially enhanced functional reactions during infection with SARS-CoV-2 [63].

Although COVID-19 was primarily considered a respiratory disorder, a significant number of patients developed other pathological conditions, including cardiovascular disorders such as myocardial damage, arrhythmia, acute coronary syndrome, and venous thromboembolism [64]. Inflammasome activation and its impacts have been documented in patients with COVID-19, hypercoagulability, and cardiovascular disorders [40,65]. Ongoing studies will be able to unveil the mechanism and linkage regarding inflammasome-induced hypercoagulopathy in leading to cardiovascular dysfunction.

## 6. Activation of the NLRP3 Inflammasome Contributed to Cardiovascular Disorders

Cardiovascular disorders are a group of disorders affecting the heart and blood vessels. It is the number one cause of death worldwide, accounting for 31% of all global deaths [66]. The role of NLRP3 activation pathways in the development of different cardiovascular disorders has been widely characterized, given their potential contribution to the development of diseases such as atherosclerosis, myocardial infarction (MI), and other cardiomyopathies [65].

Atherosclerosis is an inflammatory condition characterized by the accumulation of low-density cholesterol and large numbers of immune cells (dendritic cells and lymphocytes) in blood vessels. The overall pathogenic event is divided into four main steps: (1) EC injury, (2) accumulation of cholesterol crystals and low-density lipoproteins (LDL), (3) adhesion, migration, and differentiation of monocytes into macrophages, and, finally, (4) recruitment and proliferation of smooth muscle cells [67,68]. Macrophages, ECs, and smooth muscle cells may serve as potential activators of the inflammasome in atherosclerosis [65,69]. EC injury is the first event in the initiation of atherosclerosis caused by mechanical, chemical, environmental and/or immunological factors. NLRP3 inflammasome activation by tobacco smoke and certain environmental pollutants, such as cadmium and acrolein, results in pyroptosis and injury of ECs [70,71,72]. Moreover, cholesterol crystals are an essential factor in the induction of endothelial dysfunction in blood vessels. Cholesterol crystals can act as a DAMP signal and promote progression of atherosclerosis through interactions with immune cells. Monocytes attach to the lesions on injured ECs and differentiate into macrophages and foam cells [73]. NLRP3 present in the cytoplasmic matrix of macrophages and foam cells can initiate atherosclerosis when cholesterol crystals are engulfed by macrophages via their CD36 receptors, resulting in activation of the NLRP3 inflammasome via lysosomal damage. Early-stage atherosclerosis was reportedly reduced in an NLRP3 knockout mouse fed a high cholesterol diet, indicating a close link between activation of NLRP3 and atherosclerosis progression [74,75,76,77]. Monosodium glutamate is another DAMP signal for activation of NLRP3 and initiation of several inflammatory events, including atherosclerosis [78].

The narrowing of coronary arteries resulting from deposition of atherosclerotic plaques can lead to further complications, including MI, which is among the most common causes of death attributable to cardiovascular disorders. MI is defined as the myocardial death due to prolonged disruption of the blood supply into the heart. This situation severely affects the pumping capability of the heart, along with arrhythmia and instability in ion channel function [79,80]. Cellular injury caused by ischemic insult and subsequent release of cellular debris and metabolites can activate inflammasomes, initiating inflammation [81]. The NLRP3 inflammasome is reported to act as a primary sensor for DAMPs released during the onset of acute MI, as evident from a high expression of the inflammasome-associated proteins in cardiac fibroblasts and cardiomyocytes. Inflammasome activation stimulates the production of IL-1β and other cytokines, initiating an inflammatory response in ischemic heart. NLRP3, along with ASC proteins and caspase-1, plays a vital role in the development of MI, cardiac fibrosis, and other adverse phenomena, as demonstrated in various animal studies [78,82,83,84]. The reperfusion procedure is usually performed to control the extent of damage caused by MI, but can also inflict damage to the myocardium. Studies on animal models of ischemia/reperfusion injury revealed overexpression of NLRP3 components. Conversely, blocking of NLRP3 resulted in a reduction of infarct size [85,86].

In addition to ischemic injury, inflammasome plays a critical role in various non-ischemic cardiovascular pathologies. Cardiac remodeling is an adaptive response that involves changes in the structure and function of the heart in order to maintain cardiac function after sustaining an injury. Following ischemic insult in rats, the NLRP3 inflammasome and calcium-sensing receptors enhanced the rate of cardiac remodeling [87]. On the other hand, a lack of NLRP3 inflammasome resulted in the onset of adverse cardiac remodeling and other phenomena, such as cardiac hypertrophy and fibrosis [88]. In one experimental setup, mice were subjected to transverse aortic constriction (TAC). Expression of NLRP3 was significantly higher in the TAC group, leading to enhanced production of specific proinflammatory and profibrotic mediators, which resulted in pathologies such as cardiomyocyte hypertrophy, fibrosis, and impaired heart function [89]. In a similar study, inhibition of the NLRP3 inflammasome resulted in better survival of TAC mouse via attenuation of left ventricular hypertrophy and fibrosis [90].

Cardiac fibrosis and remodeling are the main events in the development of congestive heart failure (CHF), which is the endpoint of many cardiovascular conditions. Data from clinical trials of CHF patients support the involvement of NLRP3 inflammasome and downstream cytokines in the pathogenesis of CHF [91]. Pressure overload in the heart triggers activation of calcium/calmodulin-dependent protein kinase IIδ (CaMKIIδ). High expression of this molecule is evident in CHF while its inhibition enhances cardiac health. CaMKIIδ activates the NLRP3 inflammasome in failing heart, triggering other adverse phenomena (like inflammation and cardiac fibrosis) through recruitment of macrophages and other inflammatory cytokines such as IL-1β and IL-18. Inhibition of NLRP3 or CaMKIIδ could preserve the myocardium from further deterioration by neutralizing the pressure overload and preventing recruitment of proinflammatory cytokines [92,93,94]. Deficiency of the epigenetic regulator ten-eleven translocation 2 (Tet2) resulted in more significant cardiac dysfunction because of increased expression of IL-1β and NLRP3. Sano et al. [95] demonstrated that selective inhibition of the NLRP3 inflammasome significantly reduced the risk of heart failure development in mice. Similar or conflicting findings have been reported by other investigators [68], highlighting the necessity for more studies to fully clarify the association between the effect of NLRP3 activation and incidence of cardiovascular dysfunction.

A similar mechanism has been shown in the severely ill COVID-19 patients with cerebrovascular complications, which involved vascular and neurological abnormalities [96,97]. The NLRP3 inflammasome has been shown to be a crucial component in neurological diseases including cerebrovascular diseases. NLRP3 was activated by thioredoxin-interacting protein (TXNIP), an endogenous inhibitor of the antioxidant thioredoxin (TRX) system, thus leading to inflammation and brain tissue injury [98]. Damage of brain tissue results in cerebrovascular disorders. On the other hand, the expressions of P2X7 receptor (P2X7R), NLRP3 inflammasome components, and cleaved caspase-3 were significantly detected in the ischemic brain tissue after stroke [99]. This murine model indicated that the P2X7R/NLRP3 pathway activates neuronal apoptosis post ischemic injury [99]. Further NLRP3 activation via NF-κB and mitogen-activated protein kinase (MAPK) signaling pathways have been reported following ischemic stroke [100]. Moreover, the ACE2 receptor as the target of SARS-CoV-2 was highly expressed in neurons, astrocytes, and oligodendrocytes, which may activate inflammation through NLRP3 pathways and lead to cerebrovascular and nerve system dysfunctions [101]. More evidence is required, however, to recognize the NLRP3 inflammasome activation pathway as a potential therapeutic target for the treatment of cerebrovascular disorders in COVID-19.

## 7. Potential Role and Therapeutic Target of the NLRP3 Inflammasome for Cardiovascular Complications in COVID-19

Respiratory disorders and hypoxia, along with the inflammatory response, cause significant myocardial damage in COVID-19 patients and are highly associated with the development of heart failure [102]. Cytokine burst induced by COVID-19 results in excessive demand for metabolites and energetics, further aggravating myocardial injury [103]. The inflammatory response caused by NLRP3 inflammasome activation under particular cardiovascular conditions results in hyperinflammatory responses by intensifying the inflammatory response induced by COVID-19 or by modulating the angiotensin-converting enzyme 2 (ACE2)/angiotensin signaling pathway [104]. ACE2 is highly expressed in both lung and cardiac cells. The severity of symptoms experienced with COVID-19 cardiovascular complications may be attributed to the high expression of ACE2 in cardiac cells, and myocardial injury is potentially attributable to ACE2-related signaling pathways [104,105]. Hyperinflammation facilitates infiltration of viral particles into other organ systems like the cardiovascular system. NLRP3 is activated by sensor proteins during viral infections once PAMPs are detected by cell surface receptors such as TLR2/4 and TREM family receptors. The decreased level of ACE2 in cardiac cells leads to increased DAMP levels, thus increasing the comorbidity of SARS-CoV-2-induced cardiovascular dysfunction [103].

High mortality rates are reported in COVID-19 patients displaying an elevated expression of D-dimers and fibrin degradation products, resulting in the onset of disseminated intravascular coagulation and other coagulopathies [106,107]. Myocardial injury, mainly attributed to the ACE2 pathway, is also observed in a significant number of COVID-19 cases. Other possible mechanisms include the cytokine burst triggered as a consequence of respiratory dysfunction [108]. Arrhythmia is another commonly observed cardiovascular condition among COVID-19 patients. Large-scale cytokine expression, such as IL-1, IL-6, and TNF- α, is possibly another possible cause of arrhythmia [109]. In addition, clinical studies have shown that COVID-19 patients with cardiometabolic comorbidities are at increased risk of having severe conditions because these factors have the potential to induce hyperinflammatory conditions [104]. Adverse cardiovascular events following infection are predominantly attributed to inflammasome activity, but the degree of involvement of inflammasomes in the cardiac manifestations of COVID-19 requires further investigation using animal models [15].

Recent preclinical studies support the potential therapeutic applicability of many existing drugs in blocking NLRP3 components as targets for treating cardiovascular disorders (Table 1). The antifibrotic small-molecule drug pirfenidone has utility in inhibiting the NLRP3 inflammasome and suppressing IL-1β-induced profibrotic and proinflammatory responses in a TAC-induced mouse model of hypertension. This treatment strategy is reported to be useful in alleviating myocardial fibrosis and enhancing overall survival of TAC-induced mice [90]. Treatment with the anti-inflammatory drug rosuvastatin ameliorated adverse phenomena such as cardiac remodeling and dysfunction via silencing NLRP3, ASC and pro-caspase-1 in rats with induced type 2 diabetes [110]. The role of rosuvastatin in blocking the activation of NLRP3 inflammasome was further investigated in ECs. Simvastatin and mevastatin effectively attenuated the expression of NLRP3 inflammasomes by inhibiting oxidized low-density lipoproteins or TNF-α [111].

Antioxidative stress agents such as cinnamaldehyde and allopurinol alleviated cardiac inflammation and fibrosis by blocking NLRP3 inflammasome activation via the CD36-induced TLR4/6-IRAK4/1 signaling pathway in rats with fructose-induced metabolic syndrome and in cell models [112]. Brief treatment with colchicine, another commonly used anti-inflammatory drug, improved cardiac health by attenuating the NLRP3 inflammasome pathway and other proinflammatory cytokines. Short-term treatment reduced infarct size and was helpful in enhancing survival and cardiac function after acute MI [113]. Cholecalciferol cholesterol emulsion (CCE) is clinically used for the treatment of vitamin D deficiency and other disorders. In an earlier investigation into the potential of CCE as a treatment agent for autoimmune myocarditis, the group showed that CCE attenuated myocarditis by downregulating the pyroptosis pathway in mice [114]. Metformin is a widely used therapeutic agent for type 2 diabetes to conserve cardiac health, and exerts its effects via several pathways. In a study on diabetic cardiomyopathy, metformin induced inhibition of NLRP3 inflammasome via induction of adenosine monophosphate-activated protein kinase (AMPK)-activated autophagy [115]. In diabetic mice treated for eight weeks with the antidiabetic drug empagliflozin, pyroptosis, cardiac hypertrophy, and fibrosis were ameliorated through activation of the sGC-cGMP-PKG pathway [116]. The mechanisms and potential therapeutics targeting cardiovascular disorders in COVID-19 are illustrated (Figure 2).

These therapeutic options are shown to limit the NLRP3 activation, and the release of IL-1β—one of the proinflammatory cytokines detected during COVID-19 progression. Recent studies on MCC950, a selective NLRP3 inhibitor, have promisingly blocked IL-1β secretion after SARS-CoV-2 stimulation of patient monocytes/macrophages with spike protein [44]. MCC950 also inhibits atherosclerotic lesion development in apolipoprotein E–deficient mice via inhibiting NLRP3 [118]. Evidence shows that dexamethasone reduces allergic airway inflammation in vivo by inhibiting the function of the NLRP3 inflammasome. After treatment, the protein levels of pro-caspase-1, caspase-1, IL-1, IL-6, and IL-17 in lung tissues decreased [119]. Anakinra, a non-glycosylated recombinant antagonist targeting human IL-1 receptor, is a 17-kD protein and is given to reduce the release of IL-6 triggered in COVID-19 patients. The administration of anakinra inhibits the effect of IL-1β and further inflammation [120]. In addition, colchicine used to treat AMI has also been selected as a therapeutic option in clinical trials and in severe COVID-19 patients [113,121]. Colchicine prevents pyroptosis and NLRP3 assembly and activation [122], thereby preventing platelet aggregation, cytokine storm, and thrombosis [121]. Further research is needed to evaluate the effect of MCC950, dexamethasone, anakinra, and other potential therapeutics in the prevention of inflammasome induced thrombotic events and cardiovascular disorders during COVID-19 progression.

## 8. Conclusions

According to recent reports, about one in five people worldwide could be at increased risk of severe COVID-19 disease. Targeted drugs are not available for many cases of asymptomatic and moderate viral infections, and only supportive care is provided. Improved understanding of the pathogenesis of hypercoagulopathy in COVID-19 patients is essential to improve care and reduce morbidity and mortality rates.

Innate immune effector cells are implicated in the pathophysiological mechanisms linking thrombosis and inflammation in different viral infections. Inflammasome components and mechanisms of inflammatory activation have recently been described for COVID-19, which provide new perspectives and research opportunities for clarifying disease pathogenesis. NLRP3 has been established as a critical component of the innate immune system that detects a wide range of microbial patterns, endogenous hazard signals, and environmental irritants, mediating caspase-1 activation and secretion of IL-1β/IL-18 proinflammatory cytokines in response to microbial infection and cellular damage. Further studies are warranted to validate the role of NLRP3 in clinical outcomes of COVID-19 and the biological mechanisms involved in the inflammatory response to infection.

In addition to chemotherapeutic drugs, natural products, and herbal drugs influence the NLRP3 pathway and can be potentially applied for cardiovascular disorders [68]. While treating cardiovascular complications by targeting NLRP3 pathway components clearly provides new opportunities in drug development, further studies and clinical trials are required to validate safety and efficacy of the therapeutic compounds in COVID-19 patients.

## Figures and Tables

**Figure 1 cells-10-00916-f001:**
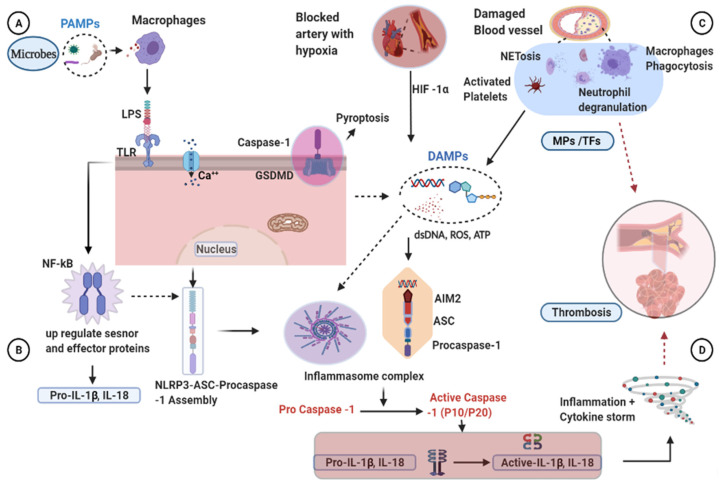
Mechanisms of inflammasome activation and clot formation. (**A**,**B**) TLR detects the presence of microbes (LPS) and activates the NF-kB that initiates the priming phase by transcribing the pro-IL-1β. Then, coupled with endogenous signals, the inflammasome components will be assembled and activate the procaspase-1. (**C**) Blood vessel damage, activated platelets, and hypoxia-inducible factors have been shown in thrombosis activation. Macrophages and neutrophils migrate and release their danger signals to the damaged vessels, including ROS, ATP, and dsDNA, contributing to the development of inflammasome complexes. Activated caspase-1 in turn activates pro-IL-1β and IL-18. Active caspase-1 further activates the GSDMD transmembrane protein associated with membrane pore formation, destabilization, and cell membrane rupture (pyroptosis). Inflammation and pyroptosis are enhanced by activated caspase-1, leading to thrombosis. AIM2, after sensing the dsDNA released from the pyroptotic cells, will form inflammatory complexes and cleave the procaspase-1 to become the active caspase-1. MPs derived from activated cells contain TFs, which initiate the coagulation system and cause blood clot formation. (**D**) Activated proinflammatory cytokines increase inflammation and cause cytokine storms and induce blood clotting. AIM2, absent in melanoma 2; ATP, adenosine triphosphate; Ca, calcium; ASC, apoptosis-associated speck-like protein containing a caspase recruitment domain; DAMPs, damage-associated molecular patterns; dsDNA, double stranded deoxyribonucleic acid; GSDMD, Gasdermin D; HIF-1α, hypoxia inducible factor one alpha; IL-1β, Interleukin 1 beta; IL-18, Interleukin18; LPS, lipopolysaccharide; MPs, microparticles; NET, neutrophil extracellular trap; NF-kB, nuclear factor kappa light chain stimulation of activated B cells; NLRP3, nucleotide-binding oligomerization domain, leucine-rich repeat, and pyrin domain containing 3 protein; PAMPs, pathogen-associated molecular patterns; ROS, reactive oxygen specious; TFs, tissue factors; TLR, toll-like receptor.

**Figure 2 cells-10-00916-f002:**
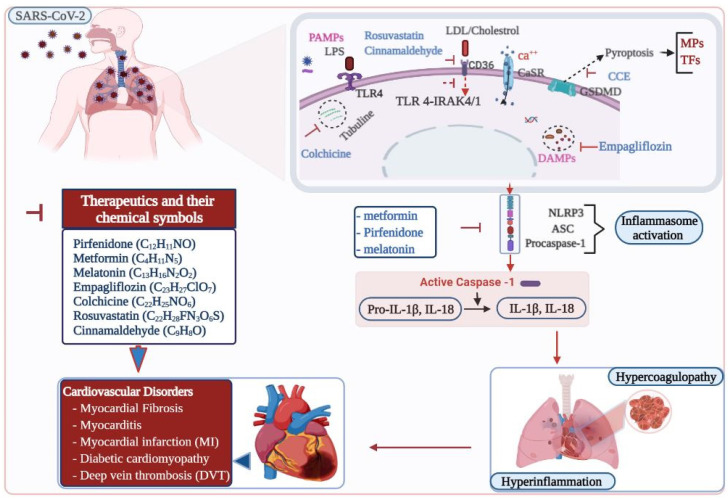
Therapeutic targets in inflammasome induced cardiovascular disorders in COVID-19. SARS-CoV-2, the causative agent for COVID-19, causes pulmonary infection. Inflammasomes are activated in SARS-CoV-2 infected patients. Activation of NLRP3 inflammasome results in hyperinflammation and hypercoagulopathy, which ultimately leads to cardiovascular complications in COVID-19. Various therapeutic compounds are indicated in inhibiting the pathways leading to the inflammasome activation and preventing cardiovascular risks. ASC, apoptosis-associated speck-like protein containing a caspase recruitment domain; Ca, calcium; CaSR, calcium-sensing receptor, CCE, cholecalciferol cholesterol emulsion; CD, cluster of differentiation; DAMPs, damage-associated molecular patterns; GSDMD, gasdermin D; HIF-1α, hypoxia inducible factor one alpha; IL-1β, Interleukin 1 beta; IL-18, Interleukin 18; IRAK, interleukin-1 receptor-associated kinase; LDL, low density lipoprotein; LPS, lipopolysaccharide; MPs, micro-particles; NLRP3, nucleotide-binding oligomerization domain, leucine-rich repeat, and pyrin domain containing 3 protein; PAMPs, pathogen-associated molecular patterns; SARS-CoV-2, severe acute respiratory syndrome coronavirus two; TFs, tissue factors; TLR 4, toll-like receptor.

**Table 1 cells-10-00916-t001:** Therapeutic compounds targeting inflammasome NLRP3 activation in cardiac disorders.

Cardiac Disorders	Therapeutic Drugs	Mechanisms of NLRP3 Inflammasome Regulation
Myocardial fibrosis	Pirfenidone	Inhibit NLRP3-induced inflammatory and profibrotic responses [90]
Cardiomyopathy	Statins, Rosuvastatin	Inhibit oxidized low-density lipoprotein or tumor necrosis factor-α [110,111]
Cardiac inflammation and fibrosis	Cinnamaldehyde	Blockage of CD36-induced TLR4/6-IRAK4/1 signaling pathway [112]
Acute myocardial infarction (AMI) myocarditis	Colchicine	Inhibition of excessive tubulin polymerization [113]
	CCE ^1^	Downregulates pyroptosis pathway [114]
Diabetic cardiomyopathy	Metformin	Activate AMPK ^2^, enhanced autophagy via inhibition of the mTOR pathway [115]
	Melatonin	Inhibiting lncRMALAT1/miR-141-mediated NLRP3 inflammasome activation and TGF-β1/Smads signaling [116]
Cardiac hypertrophy and fibrosis	Empagliflozin	Inhibition of oxidative stress-induced injury via sGC-cGMP-PKG pathway [117]

^1^ Cholecalciferol cholesterol emulsion, ^2^ Adenosine monophosphate-activated protein kinase.

## Data Availability

Not applicable.

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
