# Peer review of "Inflammasome Activation-Induced Hypercoagulopathy: Impact on Cardiovascular Dysfunction Triggered in COVID-19 Patients"

_cells, 2021, doi:10.3390/cells10040916_

Round 1

Reviewer 1 Report

Inflammasome Activation-induced Hypercoagulopathy: Impact on Cardiovascular Dysfunction Triggered in COVID-19 Patients

The purpose of the paper was to assessed host defense mechanism induced by NLRP3 inflammasome in COVID-19 patients. The paper suggests that activation of inflammasome in the host infected by SARS-CoV-2 is highly related to cytokine storm and hypercoagulopathy, which significantly contribute to the poor prognosis of COVID-19 patients.

The paper is very interesting because it investigates a very actual issue and the topic extensively address the issue. Language is correct, figures and tables are clear.

  • NLRP3 has been established as a critical component of the innate immune system that might be involved in cardiovascular complication of COVID-19. Are data about involvement of the same system in the pathogenesis of “not cardiovascular” complications of SarsCov2 infection? In particular, neurological complication has been described which may include thrombosis and ischemia leading to a poor prognosis. I would briefly include also this issue in the paper, especially cerebrovascular complication which may have similar underlying mechanism.

  • Several drugs have been widely used or proposed for COVID-19 such as steroid (mainly dexamethasone), anakinra, tocilizumab, and others. Is there any evidence that these pharmaceutical agents may affect NLRP3 inflammasome pathway?

Reviewer 2 Report

The manuscript was well written Language check required Remove figure from conclusion Add the below references Li, Y.; Ren, B.; Peng, X.; Hu, T.; Li, J.; Gong, T.; Tang, B.; Xu, X.; Zhou, X. Saliva is a non-negligible factor in the spread of COVID-19. Mol. Oral. Microbiol. 2020 Mallineni SK, Chandra Bhumireddy J, Nuvvula S. Dentistry for children during and post COVID-19 pandemic outbreak. Child Youth Serv Rev. 2021 Jan;120:105734. doi: 10.1016/j.childyouth.2020.105734. Guo, J.; Xie, H.; Liang, M.; Wu, H. COVID-19: A novel coronavirus and a novel challenge for oral healthcare. Clin. Oral. Investig. 2020.

Reviewer 3 Report

This review posits that activation of inflammasome complexes and NLRP3 in particular is a key and druggable step in the hypercoagulation induced by Sars-Cov-2 / COVID-19. There is certainly a large amount of literature cited and extensive information and description concerning the role of inflammasomes and NLRP3 in a variety of inflammatory responses. The large information-heavy sections 1-2 are adequate but not especially compelling reading. The potential role of NLRP3 in coagulation is a good section (section 3) which compiles and analyses previous studies and information in an informative manner.

Section 4 detailing activation of NLRP3 by SARS-CoV-2 and related spike protein and orf8b is good, citing the latest studies, detailing these reported interactions and importantly a study showing the effects of one of the few selective NLRP3 inhibitors, MCC950.  That hypercoagulation (sect 5) is a central part of the deleterious effects of COVID-19 is already pretty well understood I think, this section could have come earlier. The role of NLRP3 in cardiovascular disease is also well-explained and reviewed given the context.

The main issue is that many of the agents reported to inhibit NLRP3 and thus be of potential use in the context of COVID-19 and excessive coagulation are far from selective, or target NLRP3 as their primary action (section 7/8, Table 1). Many of the studies citing the effects of drugs like colchicine, allopurinol, the statins and metformin probably showed NLRP3 was inhibited incidentally rather than as the primary target of the treatment, or the main mechanism underlying the effects of these drugs. What evidence do the authors have that an NLRP3-specfici treatment, like MCC950, can ameliorate the effects of SARS-CoV2 of even pro-coagulant stimuli in general? The paucity of information here suggest the final sections are over-reaching and lack supporting evidence.

Round 2

Reviewer 2 Report

Now manuscript is in good shape

Overall looks good.